# Systematic detection of co-infection and intra-host recombination in more than 2 million global SARS-CoV-2 samples

Orsolya Anna Pipek [1], Anna Medgyes-Horváth [1] ✉, József Stéger[1], Krisztián Papp [1], Dávid Visontai[1], Marion Koopmans [2], David Nieuwenhuijse [2], Bas B. Oude Munnink [2], VEO Technical Working Group* & István Csabai[1]

Systematic monitoring of SARS-CoV-2 co-infections between different lineages and assessing the risk of intra-host recombinant emergence are crucial for forecasting viral evolution. Here we present a comprehensive analysis of more than 2 million SARS-CoV-2 raw read datasets submitted to the European COVID-19 Data Portal to identify co-infections and intra-host recombination. Co-infection was observed in 0.35% of the investigated cases. Two independent procedures were implemented to detect intra-host recombination. We show that sensitivity is predominantly determined by the density of lineage-defining mutations along the genome, thus we used an expanded list of mutually exclusive defining mutations of specific variant combinations to increase statistical power. We call attention to multiple challenges rendering recombinant detection difficult and provide guidelines for the reduction of false positives arising from chimeric sequences produced during PCR amplification. Additionally, we identify three recombination hotspots of Delta – Omicron BA.1 intra-host recombinants.

Due to the relatively high mutation rate of the SARS-CoV-2 virus, numerous variants emerged since the onset of the COVID-19 pandemic. As diverse variants from different lineages may circulate simultaneously, there is a chance of an individual being infected by two (or more) variants (strains) at the same time, defined here as co-infection. The first co-infection cases were reported as early as mid-late 2020[1,2] and since then increasing evidence suggests, that co-infections occur frequently, with an estimated rate of ~0.2–0.6% of the observed infections[3–6]. Elevated severity was observed in some case studies[7,8], but co-infection was also found in a patient with mild symptoms[9] and additional comprehensive studies report further conflicting findings concerning the course of co-infectious disease[2,4,10,11]. Recently, mainly recombinant lineages are circulating around the globe (Omicron XBB-variants) and also several recombinant lineages have been classified by pangolin[12–14]. For recombination to occur, an individual has to be

infected with two different lineages[13,15] which leads to the hypothesis that co-infection should take place frequently. This can happen in immunocompromised individuals, who have been described to be prolongedly infected with SARS-CoV-2[16], but also in the general population[4–6].

Co-infections have been identified by investigating samples with ambiguous pangolin assignment[14] results, heterozygous consensus sequence positions, or inconclusive PCR genotyping[5,10]. However, some studies from different countries (USA, France, United Arab Emirates, Brazil, Costa Rica) perform a systematic search for samples with evidence of two (or more) strains present, mainly focusing on Variants of Concern (VOCs), based on the presence of minor variant genomes (in other words the observed variant allele frequencies (AFs)) in lineage defining positions[2–4,17–20]. Nevertheless, to our knowledge, no worldwide comprehensive study was realized as of now that identifies

[1]Department of Physics of Complex Systems, ELTE Eötvös Loránd University, Pázmány P. s. 1A, Budapest 1117, Hungary. [2]Department of Viroscience, Erasmus University Medical Center, Rotterdam, Netherlands. *A list of authors and their affiliations appears at the end of the paper. ✉e-mail: horvath.anna@ttk.elte.hu

co-infection cases on a large-scale, exhaustive dataset that portrays the global trends of the disease. Through international effort, the Versatile Emerging infectious disease Observatory consortium[21] analyses and interprets genomic data from SARS-CoV-2 sequencing samples as one of its subprojects. Thanks to the worldwide research community, an enormous number of SARS-CoV-2 raw read datasets have been deposited to and are shared openly at the European Molecular Biology Laboratory - European Bioinformatics Institute (EMBL-EBI) European Nucleotide Archive (ENA)[22] European COVID-19 Data Portal[23–25]. This has allowed the processing of these raw datasets with a standardised workflow, developed by the VEO consortium, thereby reducing variability in the dataset from the use of different bioinformatic workflows. The outputs of this standardized analysis workflow[26], including mutations, their positions, and allele frequencies, among other genomic information are also made available, which allowed us to detect co-infection cases in more than 2 million samples around the world in a cohesive and reproducible manner, by analysing the major and minor variations in the samples.

The detection of co-infection in WGS data is hampered by several technical difficulties. (1) The vast majority of these data is obtained as a result of amplicon sequencing, where the applied amplification primers have a significant bias for specific genomic regions. For example, Bal et al. showed that observed alternate AFs in artificially mixed samples of specific ratios of different variant strains hardly represent the original mixture proportions[4]. (2) Even though an immense number of SARS-CoV-2 sequences have been shared publicly to date, typically only consensus sequences are provided[27]. This approach, however, poses a significant obstacle to identifying the coexistence of two or more variants at a single genomic position simultaneously. (3) The limited amount of unique lineage-defining variations and the otherwise high mutation rate of the genome causes difficulties in lineage designation. The presence of variant strains with many lineage-defining mutations (e.g. Alpha, Gamma, Mu, Omicron etc.) can be more reliably detected than that of ones with only a few unique genomic variations (e.g. Epsilon, Iota, Delta, subvariants of Omicron etc.). (4) The risk of contamination during laboratory procedures also cannot be overlooked. The accidental mixing of samples (especially during the time of emergence of a new variant) prepared for sequencing can create the effect of co-infection in the resulting dataset[28–31].

Different methods have been constructed for systematically detecting co-infection samples from WGS data, with some limitations. The manual evaluation of suspicious samples (i.e., samples with ambiguous PCR genotyping, heterozygous positions in the consensus sequence, or inconclusive lineage designation results) used in multiple case studies[5,10] is not efficient on a worldwide dataset with millions of samples. Most studies employ the basic bioinformatic pipeline of aligning sequenced short reads to a reference genome, using a variant-caller to identify mutations and finally determine the presence of co-infection from the resulting AF distributions of lineage-defining mutations. Most of these methods differ in their applied filtering criteria and the set of considered lineage-defining mutations[2,4,17,18]. A metagenomic approach using amplicon sequence variant like (ASV-like) fragments was proposed by Molina-Mora et al. to essentially separately assign each read of the sequenced sample to an appropriate lineage. Their method was tested on artificially mixed samples, but they identified no real co-infection cases in Costa Rica in their dataset[19]. Zhou et al. developed a method named Cov2Coinfect based on a hypergeometric-distribution model to assign the most likely lineages. Whenever the sum of the proportions of candidate lineages with consistently present lineage-defining mutations was approximately 100%, the sample was deemed to be a co-infection sample. They analysed more than 50,000 samples from the USA, resulting in a co-infection rate of 0.3–0.5%, however, their approach automatically rejects samples with inconsistent AFs (not summing to 100%), which may simply be due to sequencing bias of special amplification primer

systems or intra-host recombination of the lineages present concurrently[6].

Once a co-infection sample has been reliably identified, it is of interest, whether it is a simple mixture of its composing variant strains or if their genomic material has in parts been fused together (recombined), essentially creating a new, possibly transmissible variant. It is widely documented that recombination frequently occurs across Betacoronaviruses in bats and other animal hosts[32–35]. However, the frequency of recombination depends on the likelihood of dual infections, which is in part dictated by the epidemiology and ecology of infections in different hosts. For instance, the frequent occurrence of recombination has been attributed to the sharing of habitats of large numbers and different species of bats[36]. Whenever genetically distinct strains of a virus co-exist in a given patient during infection, novel strains can be produced with unique mutational landscapes via the process of recombination. Even though it is suspected that recombination is a common event during virus evolution, its traces may be difficult to detect, particularly in case of the SARS-CoV-2 pandemic where the genomic diversity is limited due to the short evolutionary history, and recombination of highly similar parent strains would cause virtually no distinguishable effect in the resulting recombinant genome. However, if the parent strains contain unique genomic variations that are highly specific to the given strain, the presence of recombinant genomes can be discovered when these mutations occur concurrently. This would be most likely during time periods when such new variants emerge. A sample obtained from an individual infected with a virus that is a product of recombination would contain a subset of the lineage-defining mutations of both parental strains as major variants (i.e. with high AFs), depending on the genomic position of the recombination breakpoint(s), thus the presence of recombination would be tractable from the consensus sequence[37]. For example, GISAID (consensus) sequences assigned to a Pango lineage whose prefix starts with an"X"[14,38] all belong to recombinant lineages (or their non-recombinant descendants), comprising about 2% of the total dataset.

However, prior to the transmission and spread of a recombinant genome, a recombination event must take place within a single host co-infected by the non-recombinant parental lineages. This leads to sequencing data in which the genomes of the parental lineages are most abundant, and recombination is only apparent in a small fraction of the short reads. Identifying traces of these in situ/incidental/intra-host/subclonal recombination events is unachievable from consensus sequences and require the examination of raw read datasets. Evidence of intra-host recombination for SARS-CoV-2 viruses has previously been demonstrated[39–41], but these studies either analysed only aggregated AF data obtained from raw sequencing results (thus detecting only indirect signs of recombination) or were limited to the examination of samples secured from a single patient. Additionally, even with an initial dataset of 10,000 randomly selected samples, co-infection is expected to occur in about 30–50 cases, allowing for the manual curation of these in search for traces of intra-host recombination, which has been the practice of previous studies[37].

Here we aim to construct a computational pipeline that first identifies SARS-CoV-2 samples exhibiting signs of co-infection from a database of more than 2 million samples collected worldwide and processed with a unified bioinformatic pipeline. The extensive information made available by sharing raw read datasets[22–25] afforded us the opportunity to then further investigate these co-infection cases in a meticulous manner to find traces of intra-host recombination. As manual inspection would have been unfeasible owing to the overwhelming volume of the data, an automated approach was implemented for intra-host recombinant detection. To this end, both the alternate AF distribution of defining mutations in the samples are examined for the presence of putative recombination breakpoints, and the raw short reads overlapping the genomic positions of multiple

lineage-defining mutations of the parental strains are interrogated for the simultaneous presence of more than one variation in these positions. Both strategies have previously been used for the identification of intra-host recombination by hand[4,5,9,17], albeit both come with limitations. We additionally discuss and emphasize the theoretical and technical difficulties hampering automated intra-host recombinant detection that should be taken into consideration to avoid artefacts.

Data files generated during the study, along with detailed computational pipelines and supplementary materials are available on GitHub at the repository github.com/csabaiBio /SARSCoV2-coinf[42] (https://doi.org/10.5281/zenodo.10057335).

## Results

### Co-infection samples in the CoVEO database

Co-infection samples were defined as samples in which a given ratio of the mutually exclusive variant defining mutations of at least two different viral strains was simultaneously present (see Methods and Supplementary Methods 1 for details). Based on the exact value of the threshold used for the required ratio of mutually exclusive variant-defining mutations, the number of detected co-infection samples varied considerably (Fig. 1a). Setting the threshold to a reasonable, but stringent 0.8, meaning that at least 80% of the mutually exclusive defining mutations of all consisting variants have to be present, resulted in a total number of 7700 co-infection samples out of the 2,172,927 good-quality samples of a human host in the CoVEO database[25,43], corresponding to an overall co-infection rate of 0.35%, which is in line with previous reports (displayed with the blue shaded area in Fig. 1a). There were 72 samples in the database that contained all mutually exclusive lineage-defining mutations of more than one variant.

The most abundant variant compositions of the identified co-infection samples were Delta – Omicron (BA.1), Alpha – Iota, Alpha – Epsilon and Alpha – Delta (Fig. 1b). This is in line with the fact, that the top five variants with the largest numbers of samples assigned to them in the database are Delta, Alpha, Omicron, Iota and Epsilon (Supplementary Fig. 1a). Furthermore, we found a near-linear relationship between the number of samples assigned to a given variant in the database and the number of co-infection cases containing the variant (Supplementary Fig. 1b), which emphasizes that results are largely influenced by the temporal and/or geographical distribution of the available sample set, calling attention to the importance of systematic worldwide surveillance. Other notable variant compositions include mixtures of two Omicron strains (Omicron (BA.2.12.1) – Omicron (BA.4)) and some combinations of more than two variants (Alpha – Epsilon – Zeta and Alpha – Epsilon – Iota). Additionally, all frequently observed variant mixtures were found in samples that were collected during the time period when worldwide incidences of the two (or more) variants were concurrently high (Fig. 1c, Supplementary Fig. 2).

To rule out obvious artefacts arising due to wet lab contamination, we collected the list of studies in which co-infection samples were found, and calculated study-specific co-infection prevalences (Supplementary Data 1). Naturally, the percentages of co-infection samples in the listed studies were usually markedly higher than the average rate of 0.35% (to compensate for the vast majority of the studies in the database that had exactly zero co-infection samples), but generally did not reach 10%. Outliers were studies with ENA accessions PRJNA817870 (prevalence 77%), PRJNA827817 (prevalence 71%), PRJNA853723 (prevalence 85%), PRJNA817806 (prevalence 44%) and, PRJNA809680 (prevalence 63%), the former three of which contain artificial mixtures of Omicron, Delta and Alpha viral variants in different ratios[4,44], while the latter two comprise preselected co-infection samples detected during the fifth wave of COVID-19 in France, between December 6, 2021 and February 27, 2022[4,5]. The fact that not all samples of these studies were identified as co-infection cases highlights the rigorous nature of our detection method that ensures that only high-quality

sequences with convincing evidence of the traces of all comprising variants are selected. Thus, our estimated co-infection rate of 0.35% best serves as a lower bound for the actual worldwide value. Naturally, the possibility of contamination cannot be completely ruled out with this method, but its probability is decreased due to the relatively even distribution of co-infection cases across different studies.

The geospatial distribution of the identified co-infection samples was for the most part fairly homogeneous, with some exceptions arising likely due to the uneven sequencing capacities of different countries (Fig. 1d).

For example, the highest prevalence of co-infection was measured in France (34%), but the total number of good-quality samples originating from France was relatively low (738) and many of them (206) were uploaded under the study accession numbers PRJNA817806, and PRJNA853723, and thus were specifically pre-selected as co-infection samples or artificial mixture of two viral variants (see above)[4,5]. In countries with more than 1000 good-quality samples (labelled in Fig. 1d), the co-infection rate varied in the 0–1.60% range, however, its value largely depends on the local sampling strategy, i.e. if the countries conducted random continuous monitoring or had preset criteria for the inclusion of specific samples in their workflows, once again highlighting the value of systematic global surveillance.

To resolve the ambiguities presented by uneven sequencing efforts and monitoring strategies, we also plotted the timeline of co-infection samples separately for each country with at least 1000 good-quality samples in the database in Supplementary Fig. 3. The results suggest that co-infection cases are detected in time ranges when both the number of available good-quality samples from the given country is sufficient and the local prevalences of at least two variants are simultaneously high.

For the investigation of how the local genetic diversity might influence co-infection rate, we added the current number of circulating variants, the current information entropy and the cumulative number of variants to the timeline figures of the two countries with the largest number of samples in the database (Supplementary Fig. 3, see Methods for details). We observed that both the number of concurrently present variants and the information entropy could serve as an indicator of the presence of co-infection, underlined by the moderate positive correlation between their value and co-infection rate in countries with a substantial number of samples (Supplementary Fig. 4).

Additionally, the possibility of identifying contaminated samples as cases of co-infection cannot be ignored. A distinctive clue for such instances could be that accidental mixing during an experimental procedure is likely to affect more than one sample simultaneously, thus co-infection cases sequenced on the same day, on the same instrument, in the same run, on the same flow cell could be potential products of contamination. However, such detailed metainformation could not be assessed for the samples in our dataset. Nevertheless, the large number of investigated samples ensures that rare events of artificial inter-sample mixing would not largely influence the results in a statistical sense.

### Traces of intra-host recombination in the AF distribution

A naive hypothesis would suggest that the alternate AFs measured at mutually exclusive defining mutations in a co-infection sample should directly reflect the variant proportions comprising the sample, i.e. a Delta – Omicron (BA.1) sample with variant ratios 80–20% (respectively) should have alternate AFs at mutually exclusive Delta-defining mutations of around 0.8 and at mutually exclusive Omicron (BA.1)-defining mutations of around 0.2. In such an ideal setup, the presence of intra-host recombinant genomes in a sample could be detectable by distinct shifts in the AFs of defining mutations at recombination breakpoints (Fig. 2a). This method has already been used for recombinant detection by manual visual inspection by Bolze et al.[17]. However,

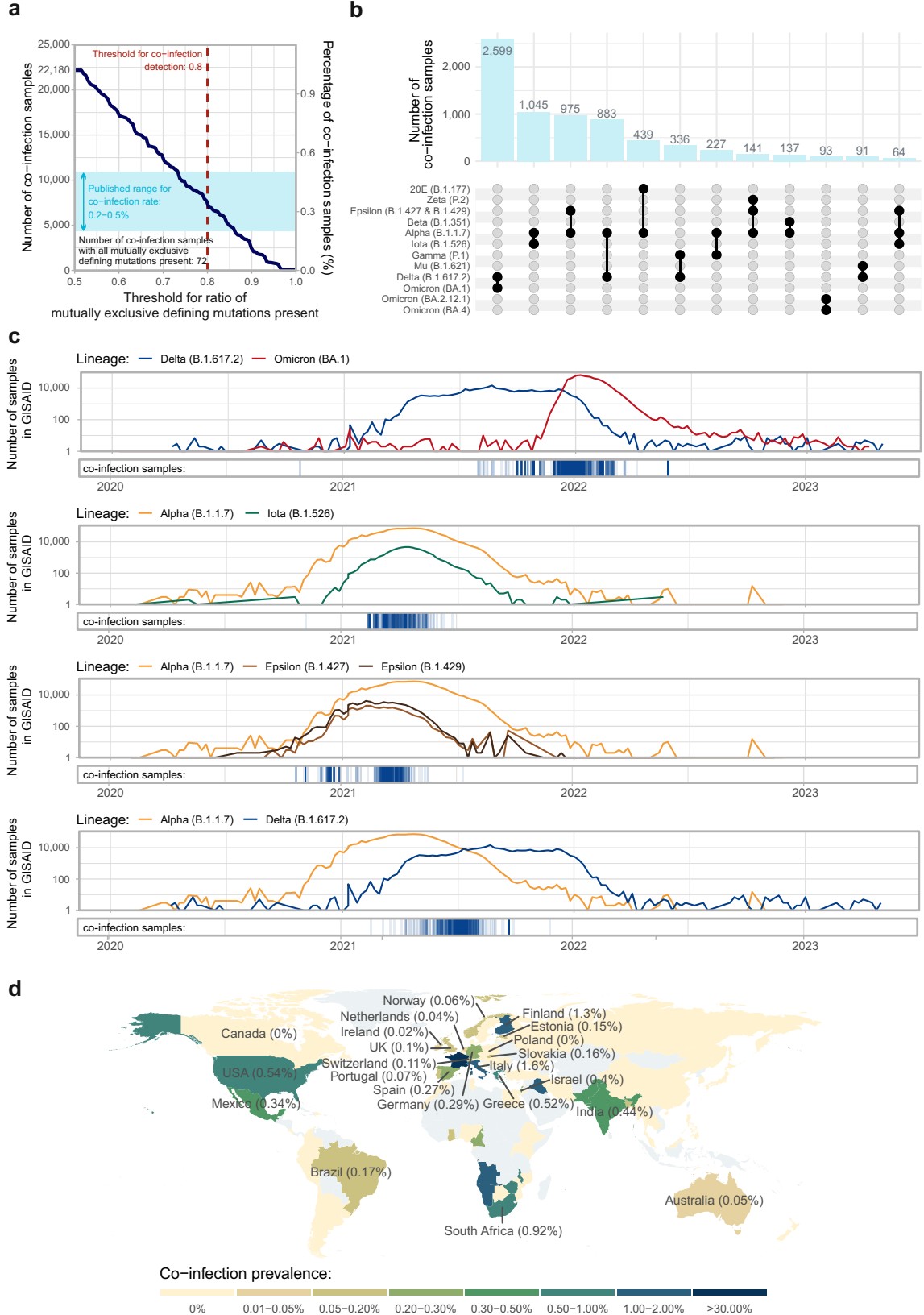

published evidence proves that measured AFs are rarely consistent with true variant proportions, mostly due to systematic bias introduced by the PCR primers used for sequencing[4]. Nevertheless, we employed a pipeline for the above-detected co-infection samples that aims to identify shifts in AFs along the genomes that could be consistent with recombination breakpoints, while simultaneously correcting for systematic offsets that were established in a previous analysis step (see Supplementary Methods 2 for the whole pipeline with detailed description). Our approach identified 13 putative recombinant samples with the ratios of recombinant genomes ranging from 6% to 16% within the samples. The majority (11) of these were Delta – Omicron (BA.1) co-infection samples, with two additional

**Fig. 1 | Co-infection samples. a** The number and percentage of co-infection samples identified in the CoVEO database with different thresholds for the required ratio of mutually exclusive defining mutations present in the variants individually. The published range for co-infection rate is indicated with the blue rectangle. The chosen threshold of the ratio of defining mutations for co-infection detection is marked with the vertical red, dashed line, corresponding to the value of 0.8, meaning that 80% of mutually exclusive defining mutations need to be present for all comprising variants individually for a sample to be considered a case of co-infection. The number of co-infection samples in which all mutually exclusive defining mutations of all comprising variants were present is 72. **b** The number of co-infection samples detected with different variant compositions. Variant strains (and Pango lineages) are listed in the order of their approximate worldwide appearance in time. Variant compositions with less than 50 co-infection samples are not shown. For a comparison with the total number of samples assigned to each strain in the CoVEO database, see Supplementary Fig. 1. **c** Temporal distribution of co-infection samples with the most frequent variant combinations. Prevalence curves indicate the number of GISAID[27] samples assigned to the respective variants (summed weekly, without including descendant lineages). Blue vertical lines on the bottom panels mark the collection date of co-infection samples of the given variants. Additional figures for variant combinations with more than 50 co-infection samples can be seen in Supplementary Fig. 2. **d** Geospatial distribution of co-infection prevalence. Countries with no good-quality samples of a human host included in the analysis are marked with light grey. Countries with more than 1000 good-quality samples of a human host are labelled, with co-infection prevalence in brackets. Co-infection prevalences in unlabelled countries are based on a low number of submitted samples and might reflect the effect of spurious data collection.

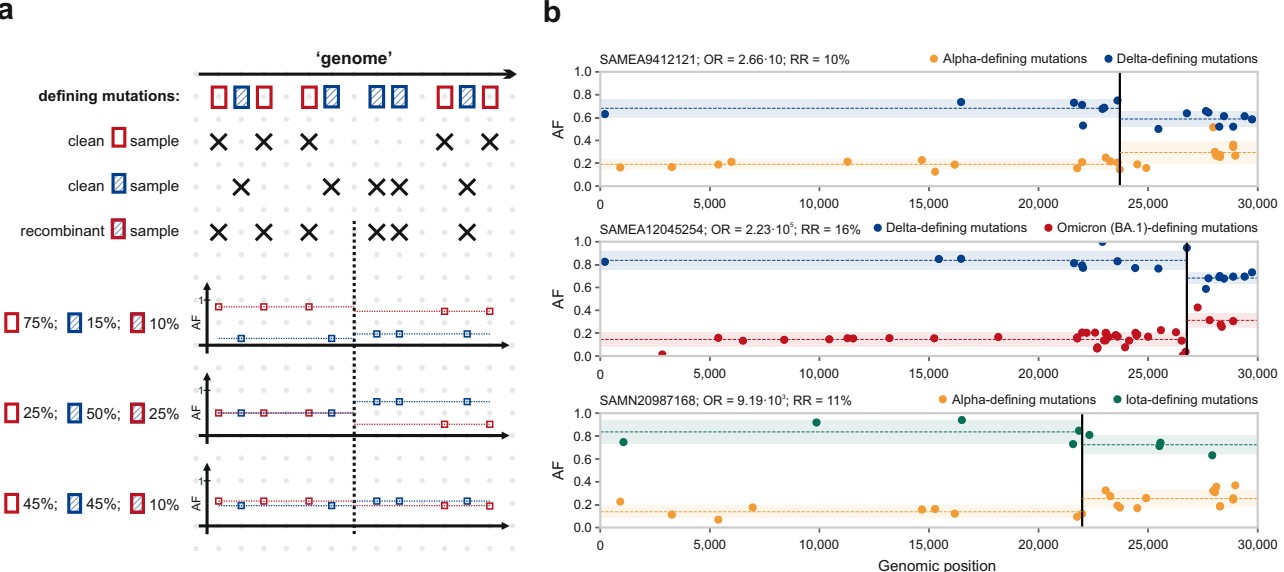

**Fig. 2 | Traces of intra-host recombination as shifts in AF. a** Theoretical AF shifts of defining mutations occurring at a recombinant breakpoint in samples containing the mixture of two parental and a recombinant strain with different ratios. **b** Examples of putative intra-host recombinant samples. (For the whole list, see Supplementary Methods 2). The black vertical line shows the location of the putative breakpoint, dashed lines and shaded, semi-transparent regions mark the means and standard deviations of relevant AFs before and after the breakpoint. OR: odds-ratio of models fitted to the data with and without a recombination breakpoint; RR: ratio of recombinant genomes in the sample.

samples of Alpha – Delta and Alpha – Iota variant combinations (Fig. 2b, Supplementary Methods 2). This is in part explained by the fact that the Delta – Omicron (BA.1) variant combination was by far the most frequent one in all co-infection samples (26%). Moreover, Omicron (BA.1)-defining mutations considerably outnumber defining mutations of other variants, thus increasing statistical power in any sort of statistical analysis.

Notably, 6 out of the 13 identified samples originated from study PRJNA817870 in which artificial Delta – Omicron mixtures were sequenced that were manually generated by mixing Delta and Omicron viral isolates in different ratios[4]. The fact that these samples were detected as putative intra-host recombinants calls attention to the ambiguities and challenges in recombinant identification, as traces of recombination is not expected in such an artificial setting. However, Supplementary Fig. S5 of Bal et al. also directs notice to the formation of chimeric consensus sequences in these samples (based on the majority rule)[4] due to the uneven distribution in measured AFs. Based on the manual inspection of Fig. S5, artificial chimera formation was present in at least 25 of the mixture samples in contrast with the 6 putative intra-host recombinants identified by our pipeline.

## Recombinant reads in raw sequencing data

Another approach for finding traces of intra-host recombination in co-infection samples is to investigate the raw reads produced during sequencing to identify ones that simultaneously carry mutually exclusive defining mutations of multiple parental strains. The whole pipeline including down-sampling of all co-infection samples, data acquisition, raw read processing, and visualization of the results can be seen in Supplementary Methods 3. During the analysis, 118 pre-selected co-infection samples were considered to decrease computation time.

Given that raw reads need to overlap mutually exclusive defining mutations of both parental strains simultaneously for this analysis, their genomic distribution is largely influenced by the distribution of mutually exclusive lineage-defining mutations. We found a significant correlation (Pearson $R = 0.725$, $p = 0.012$) between the density of mutually exclusive defining mutations and the density of reads meeting the above criteria ("overlapping reads") for different genes of the SARS-CoV-2 genome (Fig. 3a). When considering only those overlapping reads that showed signs of recombination (i.e. carried mutations of multiple parental strains), the genomic distribution of recombination breakpoints suggested by these appeared to be

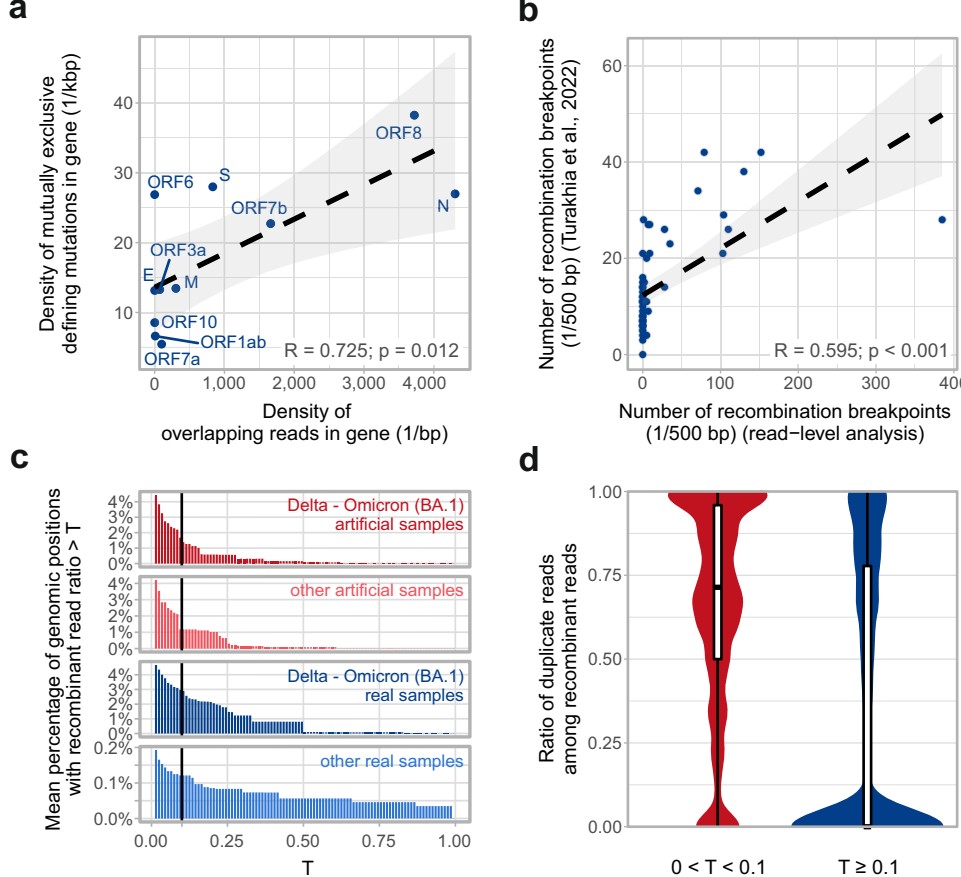

**Fig. 3 | Genomic distribution of short reads overlapping multiple mutually exclusive defining mutations and recombination breakpoints indicated by them. a** The relationship between defining mutation density and the density of reads overlapping defining mutations of multiple parental strains. All overlapping reads of the 118 analysed samples were considered for this figure. Overlapping read location is defined as the midpoint of the mutually exclusive defining mutations it overlaps. R indicates the Pearson-correlation coefficient, and the corresponding *p*-value is derived from a two-sided *t*-test ($n = 11$ genes). The dashed black line is a least-squares (linear) regression with the grey shaded area marking the 95% confidence interval. **b** The relationship between the number of recombination breakpoints indicated by overlapping reads and the number of breakpoints identified by Turakhia et al. (with the RIPPLES software) in consensus sequences[37]. Each point represents a 500 bp region of the genome. R indicates the Pearson-correlation coefficient and the corresponding *p*-value is derived from a two-sided *t*-test ($n = 60$ genomic regions). The dashed black line is a least-squares (linear) regression with the grey shaded area marking the 95% confidence interval. **c** The average percentage of genomic positions (per sample) for which the ratio of recombinant reads out of all overlapping ones reaches T (threshold). Genomic positions with exactly zero recombinant reads are not shown. Samples were categorized into groups of Delta − Omicron (BA.1) artificial/real and non-Delta − Omicron (BA.1) artificial/real samples. The vertical black line indicates T = 0.1. **d** Ratio of duplicate reads among recombinant ones for genomic positions with recombinant read ratio lower ($n = 30,491$ genomic positions across 87 samples) vs. higher ($n = 22,378$ genomic positions across 87 samples) than 0.1. Genomic positions with exactly zero recombinant reads are not shown. Violin plots and overlayed box plots demonstrate the same distributions, box edges represent the first (Q1) and third (Q3) quartiles, with the inner line showing the median value. Whiskers extend to 1.5-times the interquartile range (IQR = Q3−Q1).

moderately correlated (Pearson R = 0.595, *p* < 0.001) with that found by Turakhia et al., who analysed consensus sequences for the presence of recombination breakpoints (i.e. searched for clonal recombinants)[37] (Fig. 3b).

When exploring the average percentage of genomic positions per sample where the ratio of recombinant overlapping reads out of all overlapping reads ("recombinant read ratio") reached a given threshold of T, we found that even though artificial samples do contain a substantial number of recombinant reads, the prevalence of genomic positions overlapped by a recombinant read ratio of more than 0.1 is much lower than in true co-infection samples (Fig. 3c). This result suggests that breakpoints supported by no more than 10% of the overlapping reads might be considered artefacts due to chimera formation during PCR. This is further supported by the fact that when calculating the ratio of duplicate reads among recombinant reads in positions with a recombinant read ratio of lower vs. higher than 10%, genomic positions in which recombination was supported by less than

10% of overlapping reads show considerably higher fractions of duplicates, indicating possible evidence of PCR artefacts (Fig. 3d).

To find recombination hotspots, we considered each genomic position of the genome and calculated the number and ratio of non-artificial samples in which the given position is part of a recombination breakpoint range based on the evidence of at least 10 reads and a recombinant read ratio of at least 0.1 of all reads overlapping the genomic position (Fig. 4). We found that many of the genomic regions indicated as putative recombination breakpoint ranges in multiple samples coincide with gene boundaries. Additional intragenic hotspots were regions 22578-23202, 23525-23854, and 24130-24503 in gene S and 26530-26767 in gene M in co-infection samples of Delta − Omicron (BA.1) variants (shaded with light blue in Fig. 4). Recombination hotspots detected from short reads do not correspond to regions of recombination identified from clonal recombinants of the GISAID database[27] (samples assigned to Pango lineages XF, XS and XD, shaded with light red in Fig. 4; for more details, see Supplementary

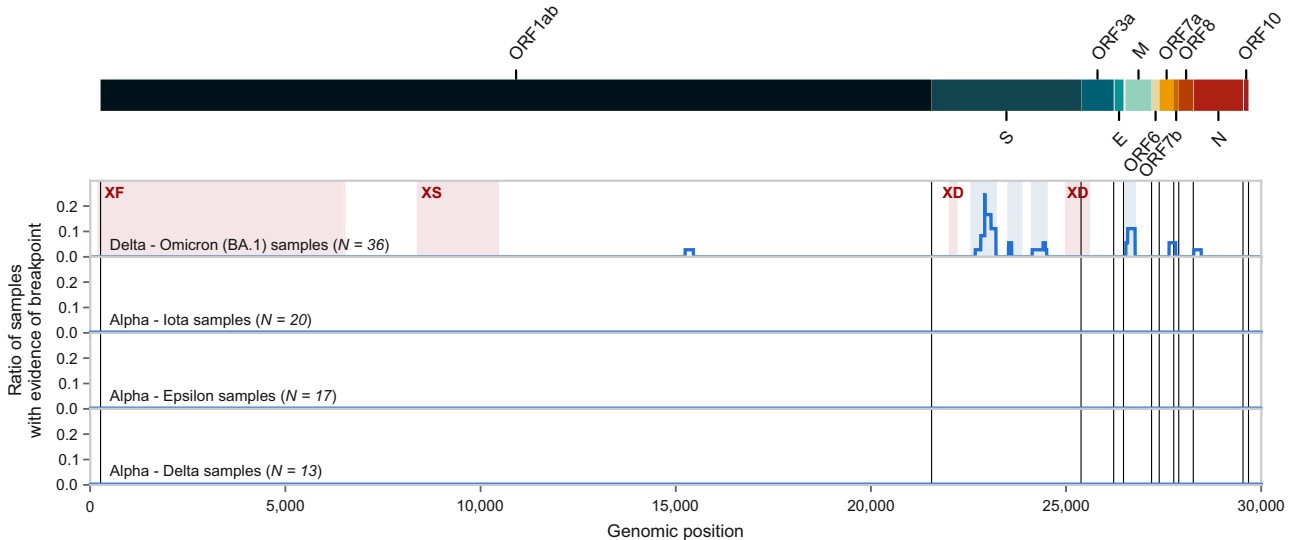

**Fig. 4 | Recombination hotspots.** The ratio of various co-infection samples with sufficient evidence of a recombination breakpoint in different genomic positions. For this analysis, non-artificial co-infection samples were categorized based on their variant composition. Only those variant compositions are shown for which at least 10 samples were available in the raw read analysis pipeline. For each genomic position, the number of samples was calculated in which the given position is part of a recombination breakpoint range based on the evidence of at least 10 short reads and a recombinant read ratio of at least 0.1 of all reads overlapping the genomic position. Regions shaded with light red are recombination breakpoint ranges of the three Pango lineages (XF, XS, and XD) in which Delta − Omicron recombination occurs (see Supplementary Fig. 5). Areas shaded with light blue indicate intragenic hotspots.

Fig. 5). None of the other frequent variant combinations had any samples with sufficient evidence of a recombination breakpoint.

We further investigated reads showing signs of recombination in non-artificial samples for the presence of genomic traces suggesting subgenomic (sg) origin (see Methods and Supplementary Methods 3) and found that recombinant reads with imprints of sgRNA were almost exclusively located at gene boundaries (S/ORF3a, E/M and ORF8/N; see Figure 7 of Supplementary Methods 3).

When analysing the 13 samples previously identified as putative intra-host recombinants, we observed no compelling evidence of recombinant reads supporting any of the presumed breakpoints. This is, in many cases, due to the lack of reads overlapping the breakpoint region. For samples with at least one overlapping read, the percentage of recombinant reads at the breakpoint position ranged from 0 to 9.5% (see Figure 8 of Supplementary Methods 3), which still falls short of the 10% threshold set to discard potential chimeric sequences.

## Discussion

The COVID-19 pandemic has brought about never-before-seen international collaboration in sample collection and high-throughput sequencing, leading to the accumulation of publicly available processed (GISAID[27]) and raw read datasets (EMBL-EBI ENA[22], European COVID-19 Data Portal[24,25]) uploaded by various laboratories and researchers worldwide. With the aim of unifying the bioinformatic processing of such diverse data sources, the VEO consortium (www.veo-europe.eu) has developed a cohesive workflow[26] that produces lists of mutations from raw sequencing data, incorporating additional key information, such as alternate AF, sequencing depth and quality scores. The CoVEO database stores this massive number of records in a queryable, searchable format, appended with detailed sample metadata, which allows for straightforward subsequent analysis. To truly harness the vast volumes of added information, compared to the more traditional approach of consensus sequence analysis, novel tools and methodologies need to be developed. In our current study, we examined more than 2 million samples with an automated pipeline for the purpose of detecting co-infection cases and traces of intra-host recombination aided by the available allele frequency and raw read information in the dataset.

The fact that we could recover a high percentage of known co-infection samples from previous studies[4,5,10,17], suggests that even with a completely independent bioinformatics workflow used for variant calling and an automated approach for co-infection detection, the results are reproducible and robust. Given the huge number of mutations in the SARS-CoV-2 genome since the onset of the pandemic, it is very difficult to assign samples to lineages based on a limited number of defining mutations, therefore we used an extended set of mutually exclusive defining mutations and very strong filtering criteria for co-infection detection, which may result in an underestimation of the number of co-infection samples, but ensures that sequencing artefacts are not identified incorrectly as signs of co-infection.

The reliability of the detected co-infection samples was further confirmed by the finding that most abundant variant combinations were the ones in which parental lineages were both separately present in large numbers in the database and that sample collection dates for co-infection samples consistently coincided with worldwide incidence peaks of their comprising variants. However, based on the near-linear trend uncovered between the number of samples assigned to a given variant and the number of co-infection samples containing that variant, the initial sample composition of the investigated dataset inherently determines the most frequent variant combinations in co-infection samples. Thus, to draw meaningful, globally relevant conclusions, the database should contain a representative sample of the worldwide cases. This underlines the value of systematic, international, synchronized surveillance and continuous sequencing.

Countries and research projects with strikingly high incidence of co-infection cases among their sequenced samples turned out to be ones specifically designed for the detailed investigation of co-infection from the outset[4,5,44]. Of note, it was previously reported that chronic infection cases, characteristic of immunocompromised individuals, contribute largely to recombination events[45], suggesting that concurrent infections with multiple variants are more likely to occur in these cases than in acute infections. Based on our results, South Africa presented one of the highest co-infection rates (0.92%), in line with the country having the largest percentage of immunosuppressed population (about 12%)[46] out of all countries with at least 1000 good-quality samples in the database.

No other dubious correlations were found between sample metadata, which could possibly indicate signs of contamination during laboratory processing. However, artefacts caused by accidental mixing, or the dropout of sequencing amplicons[47,48] can never be entirely ruled out with certainty, which accentuates the importance of providing detailed metainformation about sample preparation and sequencing procedures. Regrettably, both the recording and sharing of exhaustive metadata is generally rare and inconsistent. As an example, raw read files (FASTQ) produced by Illumina instruments contain multiple specific features of sequencing (flow cell IDs, run, lane, tile numbers, coordinates of cluster, etc.)[49], which would serve as useful guides for exploring suspicious correlations between co-infection samples. For instance, multiple co-infection samples sequenced during a single run on the same flow cell are likely to be the products of contamination. Unfortunately, given that data uploaders are free to assign generic read IDs prior to data submission, the SRA database does not store this information, as it is frequently unreliable. Nevertheless, one of the main advantages of analysing large datasets is that statistically meaningful results can be gained even in the presence of a few artificial cases.

Subsequent to co-infection detection, even more stringent selection parameters were applied for intra-host recombinant identification using multiple approaches, as several technical challenges give rise to distinct artefacts in sequencing data.

Artificially mixed samples may serve as negative controls for recombinant detection, as in purified RNA mixture 'co-infection' samples, no recombination is expected. Sovic et al. (study PRJNA827817) used RNA samples to create mixtures of two to four lineages to validate their MixviR tool[44], while Bal et al. (study PRJNA817870) worked with viral isolates from nasopharyngeal swabs inoculated on confluent Vero E6 TMPRSS2 cells and created several Delta-Omicron 'co-infection' samples, with known mixture ratios[4]. None of the samples in study PRJNA827817 were identified as putative intra-host recombinants based on AF distribution, but 6 out of the total 13 intra-host recombinants detected originated from study PRJNA817870. Bal et al. discuss in detail the emergence of artificial chimeric sequences when creating the consensus sequences of their mixture samples using the majority rule with multiple computational tools. The effect is due to the uneven distribution of AFs at defining mutations, which also causes the incorrect identification of these samples as intra-host recombinants in our pipeline. The offset in AFs is attributed to artefacts caused by primer bias during PCR amplification. Regardless, our analysis method detected only 6 of the mixture samples as putative recombinants, while the traditional approach of consensus calling resulted in chimeras for at least 25 of them (based on Supplementary Fig. S5 of Bal et al.), emphasizing that raw mutational data can provide more nuanced insights about the samples than consensus sequences. However, whenever possible, putative intra-host recombinants should be confirmed by the re-sequencing of the selected samples and hybrid capture based sequencing is preferred to reduce the risk of PCR artefacts[17].

On the other hand, based on our read-level analysis results, despite these artificial samples containing a notable number of recombinant reads, putative breakpoints supported by a recombinant read ratio of more than 0.1 is much lower than in true co-infection samples. This might indicate that using a 0.1 threshold for this parameter can be useful to avoid PCR chimera-related artefacts when identifying recombinants.

A potential mechanism of recombinant genome formation is template switching during RNA replication, i.e. RNA-dependent RNA polymerase (RdRp) switches from one RNA strand to another with different sequence during replication[50]. Template switching occurs frequently during subgenomic RNA (sgRNA) transcription, usually around transcription-regulatory sequence in body (TRS-B) sites where RdRp pauses and switches to TRS in the leader (TRS-L) sequence,

resulting in a discontinuous transcript[51,52]. Based on our read-level recombination detection on a limited subset of the co-infection samples, recombinant reads that carry genomic traces of sgRNA origin (either they contain the leader sequence and/or were soft-clipped during alignment) are indeed located exclusively near these known junction points[51]. This finding emphasizes the fact that a fraction of the detected recombination events occur during transcription and therefore have little to no effect on viral evolution. However, due to short read lengths and fairly long sequences of the transcriptome, a short read might still be sgRNA-derived, even without the presence of the leader sequence and/or soft-clipping.

The detection of intra-host recombinant genomes in co-infection samples is hampered by multiple technical and theoretical factors that together make it nigh impossible to reliably distinguish between true evidence of recombination and artefacts. Here we briefly summarize the main causes of this difficulty.

First, given that co-infection samples usually contain significant amounts of the two original parental strains, intra-host recombinant genomes comprise only a small portion of the sequenced viral population. Thus, any attempt at identifying recombinants must reckon with decreased coverage and consequently a limited amount of available data.

Secondly, well-known PCR artefacts may lead to misinterpretation of the sequencing data. PCR amplification is the standard method for the generation of sufficient genetic material prior to sequencing. Most of the sequencing data for SARS-CoV-2 has been produced by a pipeline that incorporates PCR amplification in its initial steps. Bal et al. found inconsistencies in alternate AF distribution in artificial samples of mixed variants, i.e. that alternate AFs measured at defining mutations in these samples do not correctly reflect the original mixture proportions, instead they usually overrepresent the proportion of the Delta variant[4]. Additionally, based on our observations, a systematic bias can be identified in the alternate AF distribution of defining mutations in co-infection samples of specific variant combinations. As a result of this, the presence of intra-host recombinant genomes cannot be reliably detected by subtle shifts of alternate allele frequencies along the genome. Another PCR-related artefact is the formation of chimeric sequences. It has been known for decades that during PCR amplification, PCR-mediated recombination or chimera formation systematically occurs[53], generating artificial sequences that are essentially no different from true recombinants. Even though chimera detection tools like UCHIME[54,55] are widely used in sequence analysis pipelines aiming to assess diversity or compare populations, it is virtually impossible to dependably distinguish between true recombination and chimeric sequences, and most methods simply discard all instances of chimeric/recombinant origin in the data. Generally, one must assume that chimera formation is relatively rare, while viral recombination is well-documented in laboratory settings, hence also expected to occur in co-infection samples. However, based on our estimations, putative recombination breakpoints supported by no more than 10% of the overlapping short reads are likely to be caused by chimeric sequences. By comparing the number of recombinant reads to the total number of sequenced reads in artificial samples, chimeric sequences can be present at a percentage of 0.04–0.3% or even more, emphasizing that PCR-related bias is non-negligible in sequencing data and the cautious interpretation of the results is crucial. Admittedly, there are a few experimental setups in which hybridization capture sequencing are employed, which eliminates various problematic issues introduced by amplicon-based methods[17].

Moreover, the low number and the uneven distribution of the defining mutations along the genome also inherently limit detectability. Given that, disregarding the relatively low number of defining mutations, parental strains in SARS-CoV-2 co-infection samples are highly similar, recombination events might go completely unnoticed. The identification of recombination breakpoints is constrained to the

genomic ranges between defining mutations, thus the uncertainty in their location is extremely high. Furthermore, defining mutations are unevenly distributed across genomic positions, a disproportionately high amount (considering gene lengths) of them is located on genes S and N, making it very difficult to detect recombination breakpoints occurring in other, less frequently mutated regions of the genome.

Finally, short read length also poses a challenge in recombinant detection, as direct evidence of recombination events can come from reads that simultaneously contain the defining mutations of both parental strains in a co-infection sample. This approach, however, is limited by the relatively short read lengths (100–200 bp) in sequencing data generated by Illumina platforms, as only those defining mutation pairs are overlapped by the same reads that are located close enough on the genome. The recent advances in long-read sequencing technologies (Nanopore, PacBio) might provide a solution for this problem, as they usually generate reads ranging from 10 to 100 kbp in length, though the increased sequencing error rate compared to Illumina technologies might present additional challenges[56,57] A recent study specifically utilized sequencing data generated by long-read PacBio single-molecule real-time (SMRT) sequencing technology to detect co-infection and subsequent intra-host recombination in a set of around 7000 samples from France[20]. However, the ratio of long-read sequencing data in our cohort of co-infection samples was negligible (Supplementary Fig. 6), thus the potential benefits of the technology could not be realized.

Even though co-infection detection is relatively straightforward, all the above-described obstacles render it demanding to systematically detect traces of recombination in co-infection samples with an automated pipeline. In many cases, one must resort to the manual observation of sequencing data to distinguish between artefacts and true signs of recombination, and even then, the decision usually ultimately rests on an educated guess. In our work, we methodically investigated a database containing the raw sequencing data of more than 2 million good-quality SARS-CoV-2 samples collected in the COVID-19 Data Portal[24,25,43] with worldwide sources, and reliably identified 0.35% of them as co-infection cases. We further set out to detect the presence of intra-host recombinants and employed two independent pipelines for their identification. We have shown that sensitivity is dominantly determined by the presence and density of defining mutations along the genome and that a threshold of 0.1 for the ratio of recombinant reads overlapping a given position might be reasonable to get rid of PCR-induced artefacts. Recombination hotspots were usually located at gene boundaries (with a fraction of recombinant reads carrying signs of sgRNA-origin) and three additional intergenic hotspots were identified in Delta – Omicron (BA.1) co-infection samples. Our work paves the way for further large-scale studies systematically utilizing raw read sequencing data for the detailed investigation of the recombination potential of SARS-CoV-2 in real-world, non-laboratory settings, which might help monitor and forecast important milestones in viral evolution.

## Methods

### The CoVEO database
Prefiltering steps and initial selection of co-infection samples were carried out with the use of the CoVEO database, a PostgreSQL database storing the mutational data (VCF files) of SARS-CoV-2 sequencing samples uploaded to the European COVID-19 Data Portal[24,43] (www.covid19dataportal.org). This database and the automated analyses leading to a standardised VCF file was developed by the Versatile Emerging infectious disease Observatory[21] (VEO, www.veo-europe.eu) consortium. The dataset is unique in the sense that besides the commonly available consensus sequences of the samples, it also contains low alternate allele-frequency mutations (minor variants) and sequencing depth information in a queryable format, along with sample metadata to allow for simple filtering. Additionally, samples of

the database are analysed with a standardized variant calling workflow (available on GitHub[58,59]) in order to keep technical bioinformatics artefacts at a minimum and to obtain comparable results in spite of multiple sample collectors and various laboratory protocols. Supplementary Fig. 7 demonstrates the steps of data processing applied for the creation of the analysed dataset.

### Quality filtering criteria
The altogether 3,093,454 samples of a human host in the CoVEO database were initially filtered to exclude samples that had a total base count of 100,000 or less to avoid misinterpreting sparse sequencing data. Additionally, to further ensure relatively even coverage of the viral genome, we discarded samples that had a sequencing depth of less than 10 in more than 10% of the 29,903 genomic positions of the reference genome (NCBI ID: NC_045512.2). This filtering step resulted in 2,172,927 remaining samples collected in the period from the 30th of December 2019 to the 30th of June 2022.

### Unique defining mutations of SARS-CoV-2 viral variants
Instead of using a precompiled list of genomic variations characteristic of each SARS-CoV-2 viral strain, we used the marker table[60] provided by Valieris et al.[61] in which all distinguishing mutations are listed with the number of GISAID samples containing the reference and alternate alleles for each lineage. Briefly, they treated each consensus sequence of the GISAID SARS-CoV-2 database[27,62] (Global Initiative on Sharing All Influenza Data, https://www.gisaid.org) as a single sequencing read, aligned them to the Wuhan reference sequence and called variants with the highly sensitive GATK Mutect2 tool[63].

In downstream analyses, we used the mutations listed in this table that were unique and highly indicative of specific viral strains. More precisely, for each mutation, the lineages with the largest and second-largest prevalence were identified. Genomic variations with a largest prevalence of larger than 80% and a second-largest prevalence of less than 10% were considered as "unique defining mutations" of the lineage with the highest mutational incidence. The list of unique defining mutations used in the study is listed in Supplementary Data 2. The numbers of unique defining mutations for each variant strain are shown in Supplementary Data 3.

### Identification of candidate co-infection samples
Samples of the CoVEO database that met the coverage threshold across the SARS-CoV-2 genome (see above) were considered to have moderate evidence of co-infection if more than 50% of the unique variant-defining mutations of at least two different variant strains were concurrently detected in them. No allele-frequency filtering was applied at this step to enable the identification of even trace amounts of the minor variant strain. This analysis step produced 29,666 putative co-infection samples.

### Mutually exclusive defining mutations in variant combinations
Given that the number of truly unique defining mutations was below 10 for more than half of the investigated variants (Supplementary Data 3), we extended the list of unique defining mutations by including all mutually exclusive mutations specific to each variant combination (Supplementary Data 4). For this purpose, for each variant combination indicated in any candidate co-infection sample (see above), we collected the list of mutations that were present in at least 80% of GISAID samples assigned to one of the comprising variants, while simultaneously present in less than 10% of the samples assigned to any other variant(s) of the variant combination. This way, in a hypothetical co-infection sample of Delta – Omicron (BA.2) composition, the potential number of mutually exclusive defining mutations for the Delta and Omicron (BA.2) variants is 18 and 70, respectively, even though they had only 5 and 4 truly unique defining mutations. This is explained by the fact that Omicron BA.2 shares most of its typical

mutations with various other Omicron lineages, but in a Delta – Omicron (BA.2) context these can be leveraged to differentiate between the two comprising variants. This expansion of the list of considered mutations allowed the refining of co-infection detection by largely reducing the bias introduced by using only a few defining mutations to identify samples of potentially mixed origin.

### Final selection of co-infection samples

The list of candidate co-infection samples was further filtered to only include those that carried at least 50% of the mutually exclusive variant-defining mutations of all the variants indicative of the given variant combination (see above), resulting in 22,180 potentially mixed samples.

We defined the ratio of mutually exclusive defining mutations present for a given variant as the number of mutually exclusive defining mutations found in a sample for the given variant (with a non-zero AF), divided by the potential total number of mutually exclusive defining mutations characteristic of the variant in the given variant composition. For the hypothetical Delta – Omicron (BA.2) co-infection sample of the above example this would mean, that if the sample contained 15 mutually exclusive defining mutations of the Delta, and 65 mutually exclusive defining mutations of the Omicron (BA.2) variant, the ratio of mutually exclusive defining mutations would be 0.83 (15/18) and 0.93 (65/70) for the two variants, respectively.

The number of supposed co-infection samples was then determined with multiple thresholds for the ratio of required mutually exclusive defining mutations in the range of 0.5 to 1, with the threshold applied to all consisting variants simultaneously. The final filtering limit of 0.8 was chosen for the identification of a total number of 7700 co-infection samples. This extremely stringent cut-off ensures that a co-infection sample truly contains at least trace amounts of the whole genomes of its comprising variants and inherently discards samples with only recombinant genomes (i.e. samples that can be assigned to any of the Pango lineages with a prefix starting with "X"). Theoretically, if mutually exclusive defining mutations were evenly distributed along the length of the genome and recombinants had only one breakpoint, a recombinant genome would have a ratio of mutually exclusive defining mutations of x for one of its parental variants and a ratio of 1-x for the other one, which can never simultaneously reach 0.8.

### Comparison of collection date with the prevalence of different variants

To validate that the detected co-infection samples could reasonably arise in a natural setting by the simultaneous infection of the same patient by two (or more) different viral strains, we compared the collection date of these samples with the worldwide and country level prevalence of the variants identified in them. To this end, we queried the metadata of SARS-CoV-2 samples uploaded to the GISAID website and calculated the number of samples assigned to different Pango lineages[14] for each week, then plotted their incidence curves, along with the collection dates of the samples containing those variants (Fig. 1c and Supplementary Figs. 2, 3).

### Distribution of co-infection samples

In an attempt to identify obvious giveaways of wet lab contamination and see how co-infection samples are distributed across different studies and geographical locations, we determined the number of good-quality samples of a human host with available mutation information of various study accession IDs and collecting countries in the CoVEO database. Study- and country-specific prevalence rates were calculated as the percentage of identified co-infection samples within a given study (Supplementary Data 1) or country. The details of the whole pipeline including unique and mutually exclusive defining mutation selection, along with the detection of co-infection samples can be found in Supplementary Methods 1.

### Measures of genetic diversity

The association between co-infection rate and genetic diversity was investigated through three different metrics of the latter. To calculate the number of concurrently circulating variants, simply the number of lineages with any samples assigned to them in the GISAID database collected in the given week from the given country was determined (Supplementary Figs. 3, 4). Information entropy was calculated as $S = - \sum p_i \log p_i$, where $p_i$ is the ratio of samples assigned to variant $i$ in the given week, in the given country in the GISAID database (Supplementary Figs. 3, 4). Cumulative genetic diversity was defined as the number of separate lineages present up until a time point in the country (Supplementary Fig. 3). For all these measures, linages were filtered to only include those that were accessed in our study for the presence of co-infection.

### Detection of intra-host recombinants from AF distribution

Clonally recombinant samples would in principle have genomes that were fused together from the appropriate parts of the genomes of parental viral strains at some breakpoint(s), thus would exhibit signs of different sets of mutually exclusive defining mutations of their parental variants before and after the breakpoint(s). (These are ab ovo discarded with the above co-infection detection pipeline.) During intra-host recombination, however, the two parental strains co-exist with the recombinant strain (with varying ratios) within a single sample, and putative recombinant breakpoints have previously been identified by the shifts in AFs observed for the variant-defining mutations of the parents[4,5,17]. In an ideal setting, the absolute value of the AF shift corresponds to the ratio of the recombinant genome in the sample (Fig. 2a).

To this end, we developed a pipeline that detects putative breakpoints in co-infection samples where the mean alternate AF of one set of mutually exclusive defining mutations increases, while the mean alternate AF of the other set of mutually exclusive defining mutations decreases. To remove presumed artefacts and noise, only those genomic positions were retained as possible breakpoints, that met multiple stringent filtering criteria. AFs were further corrected for systematic bias in their distribution. Odds-ratios (ORs) of a "breakpoint" and a "no-breakpoint" model were calculated for samples that had a potential breakpoint that satisfied prior requirements and ones with OR > 1 were considered to be putative intra-host recombinants. The ratio of recombinant genomes (RR) in the sample was determined from the shift in AFs at the breakpoint.

In this analysis as well, all mutually exclusive lineage-defining mutations were considered for the given variant combination to increase statistical power. To simplify the procedure, out of the 7700 co-infection samples identified above, only 7290 were examined that contained traces of exactly two variant strains.

The detailed pipeline of intra-host recombination detection from AF distributions, along with corrections for AF-bias can be found in Supplementary Methods 2.

### Detection of intra-host recombinants from short reads

As an independent analysis approach, we obtained aligned sequencing data (BAM files) of previously identified co-infection samples from ENA (European Nucleotide Archive, https://www.ebi.ac.uk/ena/) and queried them for short reads that simultaneously overlapped defining mutations of both parental strains in the sample. Reads were filtered for Phred-scale mapping quality (larger than 30) and additionally for base quality (larger than 30) at genomic positions of mutually exclusive defining mutations. If these reads concurrently carried both defining mutations, they were considered to exhibit signs of recombination. The ratio of recombinant reads out of all overlapping reads at a given genomic position was defined as "recombinant read ratio". Genomic positions that showed sufficient evidence of recombination

(at least 10 recombinant reads and a recombinant read ratio of more than 0.1) in multiple samples were considered to be recombination hotspots. The locations of these genomic ranges were compared to the recombination breakpoint ranges of GISAID samples assigned to the XF, XS, and XD Pango lineages, which are clonal recombinants of the Delta and Omicron strains (Supplementary Fig. 5). Recombinant reads were further explored for traces of subgenomic RNA by determining if they contained the nucleotides of the leader sequence utilized during transcription and/or if they were soft-clipped during alignment.

The detailed pipeline of overlapping and recombinant read identification, along with analysis results can be found in Supplementary Methods 3.

### Reporting summary

Further information on research design is available in the Nature Portfolio Reporting Summary linked to this article.

## Data availability

The analysed VCF and BAM files created with the unified pipelines[58,59] are accessible under the EBI-ENA umbrella project PRJEB45555. GISAID sequences used in this study are accessible [https://doi.org/10.55876/gis8.231020pu]. Datasets created and further processed in the current study, along with all supplementary material are available in the SARSCoV2-coinf GitHub repository at github.com/csabaiBio/SARSCoV2-coinf[42] (https://doi.org/10.5281/zenodo.10057335).

## Code availability

Analysis pipelines are available in the SARSCoV2-coinf GitHub repository at github.com/csabaiBio/SARSCoV2-coinf[42] (https://doi.org/10.5281/zenodo.10057335). The CoVEO database uses PostgreSQL version 11.12. Analyses were performed in python version 3.8.8 with the help of python modules psycopg2 (v2.9.5), pandas (v1.5.3), matplotlib (v3.3.4), numpy (v1.21.6), IPython (v7.22.0), tabulate (v0.9.0), scipy (v1.7.3) and re (v2.2.1). Aligned BAM files were manipulated with samtools[64] version 1.16. Further visualisations were created in R version 4.2.1 with R packages dplyr (v1.0.10), tidyr (v1.2.1), ggplot2 (v3.3.6), ISOweek (v0.6-2), stringr (v1.4.1), scales (v1.1.0), jsonlite (v1.8.2), ggpubr (v0.2.5), ggrepel (v0.9.1) and ComplexHeatmap (v2.12.1).

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

## Acknowledgements

We gratefully acknowledge all data contributors, i.e., the Authors and their Originating laboratories responsible for obtaining the specimens, and their Submitting laboratories for generating the genetic sequence and metadata and sharing via the GISAID Initiative, on which this research is based [https://epicov.org/epi3/epi_set/231020pu?main=true]. We are grateful to all those who uploaded and continue to upload raw read data to the EMBL-EBI European Nucleotide Archive. ORCID IDs claimed to projects analysed in this study are: https://orcid.org/0000-0001-9564-4607, https://orcid.org/0000-0001-7995-8163[65], https://orcid.org/0000-0002-5394-8896, https://orcid.org/0000-0002-0151-0657[25] The project has received funding from the European Union's Horizon 2020 research and innovation programme under grant agreements No. 874735 (VEO) (K.P., J.S., D.V., M.K., D.N., B.B.O.M., I.C.) and No.101046203 (BY-COVID) (O.A.P., A.M.-H.).

## Author contributions

O.A.P. and A.M.-H. performed the analysis and interpretation of the data and contributed to the writing of the manuscript. K.P., J.S. and D.V. contributed to the acquisition of the data. M.K., D.N. and B.B.O.M. contributed to the writing of the manuscript. I.C. contributed to the conception of the study and the writing of the manuscript. The VEO Technical Working Group provided assistance with the CoVEO database. All authors read and approved the final version of the manuscript.

## Funding

## Competing interests

The authors declare no competing interests.

## Additional information

## VEO Technical Working Group

**Guy Cochrane**[3]**, Nadim Rahman**[3]**, Carla Cummins**[3]**, David Yu Yuan**[3]**, Sandeep Selvakumar**[3]**, Milena Mansurova**[3]**, Colman O'Cathail**[3]**, Alexey Sokolov**[3]**, Ross Thorne**[3]**, Nathalie Worp**[2] **& Clara Amid**[2]

[3]European Molecular Biology Laboratory, European Bioinformatics Institute, Wellcome Genome Campus, Hinxton, Cambridge CB10 1SD, UK.

