## [Peer Review File · Nature Communications]

Systematic detection of co-infection and intra-host recombination in more than 2 million global SARS-CoV-2 samplesREVIEWER COMMENTS

Reviewer #1 (Remarks to the Author):

Piepek et al. have developed a cohesive workflow that produces lists of mutations from raw sequencing data, incorporating additional elemental information for improved identifying intra-host recombination utilized the public COVID-19 database. In this manuscript, they utilized more than 2 million samples and subjected the selected/ quantified samples to be examined with an automated pipeline for the purpose of detecting co-infection cases and traces of intra-host recombination aided by the available allele frequency and raw read information in the dataset. First, they confirmed that their pipeline also detected the published co-infection samples from previous studies, supporting an independent bioinformatics workflow including them are reproductive. Rational geographical distribution of co-infection prevalence was calculated and showed a range from 0 % to 1.6 % depending on the sample setting and circulation in the individual countries. Traces of intra-host recombination of co-infection samples were investigated. They identified the breakpoints of intra-host recombination but they were unable to calculate breakpoints occurred in a real-world because of the short read-out sequencing data. They claimed breakpoints supported by no more than 10% of the overlapping reads might be considered artifacts due to chimera formation during PCR. Overall, the methodology shown in this study is rational and well explained but it would be difficult to find the novelty and advantages from the previous studies. In addition, I would wonder how bioinformatic workflow including the authors are able to take account of and distinguish genetic-drift under immune pressure by people acquiring humoral immunity on SARS-CoV-2.

1. Figure 1: D, Could you please provide further information about more than 1000 good-quality samples? Also, it would be helpful if they could provide how prevalence of COVID-19 in the respective countries. It would be doubtful there is no intra-host recombination events happened in Canada. Please discuss.
2. Line 39: SARS-CoV-2 possess the proofreading activity so that virus itself does not cause error compared to other RNA viruses. Please rephrase.
3. Line 391: 'huge' number of mutations are mostly located in the spike proteins because people are experienced and/or vaccinated against COVID-19. I would suggest the authors to explain the relevance of breakpoints observed in the spike protein mediated by immune-escape
4. Lines 592, 596, 610: quadrangle (convention errors). Please amend

Reviewer #2 (Remarks to the Author):

This study by Pipek et al comprehensively analyzed 2 million raw reads data, to investigate the occurrence of co-infection events in the cases infected by SARS-CoV-2. The investigators defined a set of robust and standard pipeline for processing the raw read data and detecting co-infections. These methods are informative and useful for future research in this field. Most importantly, the study reports the estimated proportion of co-infection to be 0.35% among the investigated cases, providing valuable reference for profiling multiplicity of infection. I also appreciate the discussion on the challenge of analyzing intra-host raw reads data, clearly laying out the major obstacles in studying recombination using real world deep sequencing data intra-host.

Overall, the paper is nicely written and clear. I have only a few major comments:

- It is not very clear if the co-infection detection considers only co-infection by two genetic variants? Have the authors observe co-infection by three or more variants, what if the detection standard is lower to, say 50%?
- Usually it is perceived that genetic diversity would relate to recombination event. In this study, genetic diversity is not reported. I wonder if there might be a correlation between the two indicated from the data. Using each region's genetic diversity against the region's coinfection rate can easily generate such an output.
- It was reported that recombination event is prone to occur in the chronic infection cases, whilst limited among the acute infections (Xue et al 2018, trends in Microbiology, "within-host evolution of human influenza virus"). It is likely that most of the available SARS-CoV-2 genome are from

acute infections (?). Meanwhile, South Africa, which is among the region with the highest co-infection rate, has perhaps the largest immunocompromised population that may contribute to the establishment of recombinant genetic variants. These factors may be added to the discussion.

- At the macro-scale, epidemiology of SARS-CoV-2 variants show that one novel genetic variants may quickly reach dominance within 1-2 months, such as the delta and omicron. Therefore, I wonder if this phenomenon increases or decreases the genetic diversity – perhaps at global scale over all time periods, add to genetic diversity, whereas in a given region and time, lower the genetic diversity, particular after the swiping of a certain variant. Co-infection event may appear in the transition time between two major variants, however, the Figure 1 C overlays multiple regions, therefore, the transition period between predominant variants are prolonged and not obvious.

Minor comments:

- Figure 1a: “blue rectangle” – lack legend, not obvious

- Figure 1a: horizontal line at 72, not good visualization, may consider using scaled and piecewise axis.

Reviewer #3 (Remarks to the Author):

Pipek et. al Systematic detection of co-infection and intra-host recombination in more than 2 million global SARS-CoV-2 samples

The authors utilize more than 2 million raw SARS-CoV-2 sequencing data files to investigate co-infection and recombination events. Mounting evidence suggests that salutation evolution is at least partly driven by intra-host recombination events. Therefore, detecting and quantifying the extend of intrahost evolution is key to understanding the drivers of rapid SARS-CoV-2 evolution which would likely impact current mitigation strategies.

The authors systematically examine an impressive number of genomes, using raw sequencing data submitted by laboratories throughout the world. The manuscript is clear and well written with significant details of the methodology used. The authors clearly outline the limitations of the analysis conducted.

Major Comments

The authors have conducted an impressive analysis of the countries providing SARS-CoV-2 data that enabled the analysis presented. However, there is no presentation of the breakdown of sequencing technologies, or viral enrichment methodologies used. Could the authors outline at a minimum the distribution of sequencing technologies used to generate the dataset.

Further to this point is there any association between detection of Intrahost recombination events within reads and the sequencing technology used? As the authors suggest long read sequencing methodologies may provide an advantage as the longer read lengths are more likely to contain mutations from both parental strains.

More detailed information about the sequencing technology, enrichment methodology, sequencing coverage, sequencing depth, duplication rates and quality metrics would be useful. These quality control parameters are particularly important to validate results from the small number of samples where recombinant reads were detected.

Minor Comments

A simple diagram to illustrate the bioinformatic software used by the CoVEO database would be useful to enable the approach to be reproduced.

Systematic detection of co-infection and intra-host recombination in more than 2 million global SARS-CoV-2 samples

Orsolya Anna Pipek, Anna Medgyes-Horváth, József Stéger, Krisztián Papp, Dávid Visontai, Marion Koopmans, David Nieuwenhuijse, Bas B. Oude Munnink, VEO Technical Working Group, István Csabai

General remarks

We greatly appreciate the insights of all Reviewers and feel that their useful proposals helped refine the results presented in the manuscript.

Based on the various valuable suggestions, four further supplementary figures have been added to the manuscript. To keep the text coherent, already existing supplementary figures have been renumbered to accommodate the newly generated ones.

Moreover, Additional Datafile 4, uploaded to the repository github.com/csabaiBio/SARSCoV2-coinf/ have been updated with multiple fields describing laboratory procedures during sequencing and quality metrics of the resulting data (see answers to Reviewer #3).

All modifications in the manuscript are highlighted in yellow to allow for simplified tracking.

Point-by-point response to the reviewers' comments

Reviewer #1 (Remarks to the Author):

Piepek et al. have developed a cohesive workflow that produces lists of mutations from raw sequencing data, incorporating additional elemental information for improved identifying intra-host recombination utilized the public COVID-19 database. In this manuscript, they utilized more than 2 million samples and subjected the selected/ quantified samples to be examined with an automated pipeline for the purpose of detecting co-infection cases and traces of intra-host recombination aided by the available allele frequency and raw read information in the dataset. First, they confirmed that their pipeline also detected the published co-infection samples from previous studies, supporting an independent bioinformatics workflow including them are reproductive. Rational geographical distribution of co-infection prevalence was calculated and showed a range from 0 % to 1.6 % depending on the sample setting and circulation in the individual courtiers. Traces of intra-host recombination of co-infection samples were investigated. They identified the breakpoints of intra-host recombination but they were unable to calculate breakpoints occurred in a real-world because of the short read-out sequencing data. They claimed breakpoints supported by no more than 10% of the overlapping reads might be considered artifacts due to chimera formation during PCR. Overall, the methodology shown in this study is rational

and well explained but it would be difficult to find the novelty and advantages from the previous studies. In addition, I would wonder how bioinformatic workflow including the authors are able to take account of and distinguish genetic-drift under immune pressure by people acquiring humoral immunity on SARS-CoV-2.

1. Figure 1: D, Could you please provide further information about more than 1000 good-quality samples? Also, it would be helpful if they could provide how prevalence of COVID-19 in the respective countries. It would be doubtful there is no intra-host recombination events happened in Canada. Please discuss.

We thank the Reviewer for suggesting taking a closer look at local prevalence curves and timelines in individual countries. The results of this analysis are now included in Supplementary Figure 3. for countries in which the total number of good-quality samples in the CoVEO database reached 1,000. The top panel on each subfigure depicts the number of samples in the CoVEO database collected each week from the appropriate country. The second panel shows the collection date of co-infection samples identified in the given country. The third panel demonstrates the country-wise weekly prevalence curves of variants in the GISAID database that were included in the study. Coloured curves correspond to the most abundant variants in the whole set of detected co-infection samples (Delta, Omicron (BA.1), Alpha, Iota, Epsilon, 20E), while different shades of grey demonstrate the remaining lineages. For all countries, the peaks in the temporal distribution of co-infection samples align well with time ranges in which multiple variants were concurrently present in large numbers. Specifically for Canada, most of the good-quality samples in the CoVEO database were collected in 2020, while variants investigated in our study gained prevalence only around the beginning of 2021. This explains the fact that no co-infection samples were identified in the country, even though the total number of good-quality samples (4,309) would suggest an expected number of around 15 based on the average global co-infection rate. It is possible, that even though the sequencing efforts in a given country are consistent throughout the course of the pandemic, the practice of sharing raw read datasets is maintained only for shorter periods of time. Generally, co-infection cases are detected in cases when the number of available good-quality samples in the CoVEO database is sufficient and the local prevalences of at least two variants are simultaneously high.

2. Line 39: SARS-CoV-2 possess the proofreading activity so that virus itself does not cause error compared to other RNA viruses. Please rephrase.

We are grateful to the Reviewer for calling our attention to this inaccuracy. The referenced sentence is now rephrased in the text.

3. Line 391: 'huge' number of mutations are mostly located in the spike proteins because people are experienced and/or vaccinated against COVID-19. I would suggest the authors to explain the relevance of breakpoints observed in the spike protein mediated by immune-escape

We thank the Reviewer for raising the question of immune escape in the context of recombination. We certainly agree that recombinant genomes circulating in large numbers are likely to be superior in fitness to their parental strains, let that be increased immune escape or transmissibility, etc. However, there is no compelling reason to assume that intra-host recombination is purely driven by fitness optimization. We speculate that intra-host recombinants are generated randomly and, in most cases, comprise only trace amounts of the viral population in a single patient at a given time point, thus their advantage regarding immune escape is ambiguous. On the other hand, whenever such an intra-host recombinant gains dominance within an infected individual and consequently spreads to others, it is plausible that it possesses features that render it more advantageous than its parental strains. Regardless, given the lack of available information on patient follow-up and subsequent transmission events, we had no way of monitoring such rare cases.

As discussed in the manuscript, the location of recombination breakpoints is principally determined by the resolution of the detection method, i.e., the genome-wide distribution of mutually exclusive defining mutations of the parental strains. Thus, any observable trends in recombination hotspots most likely reflect the underlying defining mutation density and are not necessarily related to immune escape.

Nonetheless, amino acid resolution data regarding immune escape is available for the receptor binding domain (RBD) of the spike protein owing to various experimental and computational efforts¹⁻⁵. In these studies, immune escape from a specific antibody has been quantified as "plasma escape fraction" (ranging from 0 to 1), the fraction of yeast cells expressing the RBD with a given mutation that fell into the escape bin during fluorescence-activated cell sorting out of all such cells. Based on Figure 1. of the review article of Carabelli et al.⁶ and the deep mutational scanning data provided in Table S3. of Greaney et al.¹, the only Delta-defining mutation in the RBD (L452R) seems to cause an increased plasma escape fraction and was found in most of the investigated Delta-Omicron co-infection samples in our study that had recombination breakpoints supported by at least 100 overlapping reads and a recombinant read ratio of at least 0.1 (see Revision Figure 1. below). On the other hand, Omicron BA.1-defining mutations G446S and E484A also result in relatively high plasma escape fractions, yet neither of them was found in any of the recombinant reads of our samples. Additionally, a recent effort by Bloom and Neher⁷ aimed to estimate the fitness effects of mutations by quantifying how their actual frequency in worldwide sequencing data compares to their expected prevalence based on the underlying neutral mutation rate. In the limited amount of data we had, we could observe no dominant tendencies that would suggest that intra-host recombination is primarily driven by fitness maximization (see Revision Figure 1. below).

Given that most of the co-infection samples of our study were sequenced on Illumina instruments (see our response to Reviewer #3 below) and consequently had read lengths of a few hundred nucleobases, read-level analysis was restricted to searching for pairs of defining mutations located near enough on the genome so that they could be covered by a single read. The resulting recombinant reads simultaneously carrying markers of both parental strains were generally too short to draw meaningful conclusions about their potential advantage in fitness compared to the non-recombinant genomes. For such inquiries, long-read sequencing and increased coverage

would potentially be preferable aided by the results of deep mutational scanning regarding both plasma escape and fitness.

Revision Figure 1. Delta and Omicron (BA.1) defining mutations in the receptor binding domain of the S gene. Columns represent mutually exclusive defining mutations of the Delta and Omicron BA.1 lineages in the RBD (labels correspond to “AA change (nucleotide change)”). Fitness values of each specific mutation have been estimated by Bloom and Neher⁷. Escape values are either the mean or the maximum plasma escape fractions measured by deep mutational scanning¹ for each mutation. The bottom panel illustrates the recombination breakpoints found in the RBDs in co-infection samples, supported by at least 100 overlapping reads and a recombinant read ratio of at least 0.1. Due to short read lengths, recombinant reads did not cover the whole genomic region of the RBD, thus it was assumed that only a single breakpoint is present.

4. Lines 592, 596, 610: quadrangle (convention errors). Please amend

We greatly appreciate the thoroughness of the Reviewer and thank him/her for noticing this oversight. This is now corrected throughout the manuscript.

Reviewer #2 (Remarks to the Author):

This study by Pipek et al comprehensively analyzed 2 million raw reads data, to investigate the occurrence of co-infection events in the cases infected by SARS-CoV-2. The investigators defined a set of robust and standard pipeline for processing the raw read data and detecting co-infections. These methods are informative and useful for future research in this field. Most importantly, the study reports the estimated proportion of co-infection to be 0.35% among the investigated cases, providing valuable reference for profiling multiplicity of infection. I also appreciate the discussion

on the challenge of analyzing intra-host raw reads data, clearly laying out the major obstacles in studying recombination using real world deep sequencing data intra-host.

Overall, the paper is nicely written and clear. I have only a few major comments:

- It is not very clear if the co-infection detection considers only co-infection by two genetic variants? Have the authors observe co-infection by three or more variants, what if the detection standard is lower to, say 50%?

We thank the Reviewer for pointing out the lack of emphasis on this issue. We have indeed observed co-infection cases of more than two variants, these are marked with three filled black dots on Figure 1b. Most common variant combinations for more than two variants were Alpha – Epsilon – Zeta (141 samples) and Alpha – Epsilon – Iota (64 samples). Altogether, out of the total 7,700 co-infection cases, there were 410 samples with more than two comprising variants, 391 of these contained traces of three, 18 of four and a single sample of five variants. Revision Table 1. shows the distribution of these among specific variant combinations.

Revision Table 1. Variant combinations in co-infection samples of more than two variants. (N: total number of co-infection samples with more than two comprising variants.)

variant combination	number of co-infection samples (N = 410)
Alpha (B.1.1.7), Epsilon (B.1.427 & B.1.429), Zeta (P.2)	141
Alpha (B.1.1.7), Epsilon (B.1.427 & B.1.429), Iota (B.1.526)	64
Alpha (B.1.1.7), B.1.623, Iota (B.1.526)	40
Alpha (B.1.1.7), Epsilon (B.1.427 & B.1.429), Gamma (P.1)	26
Alpha (B.1.1.7), Gamma (P.1), Iota (B.1.526)	24
Alpha (B.1.1.7), B.1.177, Beta (B.1.351)	21
Alpha (B.1.1.7), Delta (B.1.617.2), Gamma (P.1)	18
Alpha (B.1.1.7), B.1.1.318, Iota (B.1.526)	18
Alpha (B.1.1.7), Delta (B.1.617.2), Mu (B.1.621)	8
Alpha (B.1.1.7), Delta (B.1.617.2), Epsilon (B.1.427 & B.1.429), Gamma (P.1)	5
Alpha (B.1.1.7), B.1.1.318, Epsilon (B.1.427 & B.1.429), Iota (B.1.526)	5
Alpha (B.1.1.7), Delta (B.1.617.2), Iota (B.1.526)	5
Alpha (B.1.1.7), Delta (B.1.617.2), Lambda (C.37), Mu (B.1.621)	4
Alpha (B.1.1.7), B.1.623, Epsilon (B.1.427 & B.1.429)	3
Alpha (B.1.1.7), B.1.623, Epsilon (B.1.427 & B.1.429), Iota (B.1.526)	3
Alpha (B.1.1.7), B.1.1.318, Delta (B.1.617.2)	3
Alpha (B.1.1.7), B.1.1.318, Gamma (P.1)	2
Alpha (B.1.1.7), B.1.177, Delta (B.1.617.2)	2
Alpha (B.1.1.7), Beta (B.1.351), Epsilon (B.1.427 & B.1.429)	2
Alpha (B.1.1.7), Epsilon (B.1.427 & B.1.429), Lambda (C.37)	2
Alpha (B.1.1.7), B.1.1.318, Epsilon (B.1.427 & B.1.429)	2
Alpha (B.1.1.7), Iota (B.1.526), Kappa (B.1.617.1)	1
A.23.1, Alpha (B.1.1.7), Beta (B.1.351)	1

Delta (B.1.617.2), Gamma (P.1), Mu (B.1.621)	1
Alpha (B.1.1.7), B.1.177, Gamma (P.1)	1
Alpha (B.1.1.7), Delta (B.1.617.2), Lambda (C.37)	1
Alpha (B.1.1.7), B.1.1.318, Gamma (P.1), Iota (B.1.526)	1
Alpha (B.1.1.7), Lambda (C.37), Mu (B.1.621)	1
Alpha (B.1.1.7), B.1.1.318, Epsilon (B.1.427 & B.1.429), Gamma (P.1), Iota (B.1.526)	1
Alpha (B.1.1.7), Eta (B.1.525), Iota (B.1.526)	1
Alpha (B.1.1.7), B.1.177, Epsilon (B.1.427 & B.1.429)	1
Alpha (B.1.1.7), Delta (B.1.617.2), Omicron (BA.1)	1
Delta (B.1.617.2), Omicron (BA.1), Omicron (BA.2.12.1)	1

To further assess how the detection threshold influences the number and type of co-infection samples with multiple (more than two) variants, we re-created Figure 1a of the manuscript in Revision Figure 2. with a specific focus on multi-variant co-infection samples.

Revision Figure 2. Co-infection samples with more than two comprising variants. The number of multi-variant co-infection samples (blue) and their percentage among all co-infection samples (red) are shown in the function of the detection threshold set for the ratio of mutually exclusive defining mutations present. Pie charts on insets indicate the distribution of variant compositions in multi-variant co-infection samples with the given detection threshold (0.625, 0.725 and 0.825, respectively).

Naturally, increasing the detection threshold leads to fewer multi-variant co-infection samples. Additionally, their percentage among all co-infection samples also decreases, because a larger number of comprising variants results in a reduced probability of all of them simultaneously meeting the detection criteria. The inset pie charts in the figure demonstrate the distribution of variant compositions observed with different arbitrary detection thresholds (0.625, 0.725 and 0.825, respectively). Most common combinations were Alpha – Epsilon – Zeta, Alpha – Epsilon – Iota, Alpha – Delta – Gamma and Alpha – B.1.623 – Iota, which did not seem to be greatly affected by the detection threshold, although a stricter filtering increased the abundance of Alpha – Epsilon – Iota and Alpha – B.1.623 – Iota samples compared to Alpha – Epsilon – Zeta ones. Additionally, the ratio of samples with variant compositions including more than three variants decreased with increasing threshold values.

- Usually it is perceived that genetic diversity would relate to recombination event. In this study, genetic diversity is not reported. I wonder if there might be a correlation between the two indicated from the data. Using each region's genetic diversity against the region's coinfection rate can easily generate such an output.

We thank the Reviewer for raising this very relevant point. We opted to use either the number of concurrently present variants or the information entropy as the measure of (time-dependent) genetic diversity (Supplementary Figure 4.). For the former, we simply calculated the number of lineages with any samples assigned to them in the GISAID database collected in the given week from the given country (left panels, horizontal axes on Supplementary Figure 4). (Lineages were filtered to only include those that were accessed in our study for the presence of co-infection.) Weekly co-infection rate was determined as the percentage of co-infection samples among good-quality samples in the CoVEO database collected in the given country, in the given week (vertical axes, all panels). Each marker in the figures represents the data from a single week. Weeks during which less than 10 good-quality samples were collected were excluded from the figure. Marker size corresponds to the number of good-quality samples available. On the horizontal axes of the right panels, information entropy was calculated as $S = -\sum p_i \log p_i$, where p_i is the ratio of samples assigned to variant i in the given week, in the given country in the GISAID database. In this case as well, lineages were restricted to the ones included in the co-infection detection pipeline. “R” and “p” indicate Pearson-correlation coefficients and respective p-values (non-significant ($p \geq 0.05$) results are not displayed in the figure). In countries, where a substantial number of good-quality samples were available (United States, United Kingdom) in the CoVEO database, a moderate positive correlation was found between co-infection rate and genetic diversity (both the number of concurrent lineages and entropy). A similar trend was observable for other countries as well, although with a somewhat diminished statistical significance, most likely due to the decreased number of samples.

- It was reported that recombination event is prone to occur in the chronic infection cases, whilst limited among the acute infections (Xue et al 2018, trends in Microbiology, “within-host evolution

of human influenza virus”). It is likely that most of the available SARS-CoV-2 genome are from acute infections (?). Meanwhile, South Africa, which is among the region with the highest co-infection rate, has perhaps the largest immunocompromised population that may contribute to the establishment of recombinant genetic variants. These factors may be added to the discussion.

We thank the Reviewer for proposing this idea. Although the amount of metadata on viral hosts is generally limited, we also assume that a significant portion of the available SARS-CoV-2 genomes originate from acute infections. To evaluate whether the ratio of the immunocompromised population within a country can relate to the co-infection rate, we utilized the dataset provided by Clark et al.⁸ Conditions “cancers with direct immunosuppression”, “cancers with possible immunosuppression” and “HIV/AIDS” were considered immunocompromising, and their relative country-wise prevalence was summed for each country (essentially assuming that the three diseases were mutually exclusive). The relationship between the percentage of immunocompromised populations and co-infection rate is depicted in Revision Figure 3.

Revision Figure 3. Relationship between the percentage of immunocompromised populations and co-infection rate. Each dot represents a single country with the marker size corresponding to the number of good-quality samples originating from the given country in the CoVEO database. Immunocompromised population was defined as the number of patients with either cancers with direct or possible immunosuppression or HIV/AIDS patients (data from Clark et al.⁸). Countries with less than 1,000 good-quality samples are not shown.

South Africa is a remarkable outlier in the figure, both in terms of its immunocompromised population and co-infection rate, although no convincing correlation was observed between the two variables for the rest of the countries. However, based on this result, we agree with the Reviewer that the unusually high number of co-infection cases for South Africa might indeed be related to the high incidence of immunocompromising conditions in the country. This is now also briefly discussed in the manuscript. Additional data regarding the number of untreated HIV/AIDS

cases in each country would allow for the further fine-tuning of this analysis, given that patients undergoing treatment are less likely to be at increased risk of prolonged infections from SARS-CoV-2.

- At the macro-scale, epidemiology of SARS-CoV-2 variants show that one novel genetic variants may quickly reach dominance within 1-2 months, such as the delta and omicron. Therefore, I wonder if this phenomenon increases or decreases the genetic diversity – perhaps at global scale over all time periods, add to genetic diversity, whereas in a given region and time, lower the genetic diversity, particular after the swiping of a certain variant. Co-infection event may appear in the transition time between two major variants, however, the Figure 1 C overlays multiple regions, therefore, the transition period between predominant variants are prolonged and not obvious.

We thank the Reviewer for his/her line of reasoning, we have now included a separate timeline of co-infection events and prevalence curves for each country with a total number of good-quality samples of at least 1,000 in Supplementary Figure 3 (see detailed description in legend and in our answer to the first question of Reviewer #1).

As an additional insight, we included a further panel depicting cumulative genetic diversity (defined as the number of separate lineages present up until a time point in the country) and (current or time-dependent) genetic diversity (defined either as the number of concurrently present lineages or the information entropy, see our answer above) binned weekly for the two countries with the largest numbers of samples in our database (United States and United Kingdom; Supplementary Figure 3).

Based on our observations, the appearance and the consequent dominance gaining of a new variant can best be monitored by investigating temporally local peaks in entropy. With the emergence of a new variant, the previously monotonous landscape (i.e. a zero-entropy phase) of prevailing lineages shifts and entropy increases up until both the previously pervading and the newly spreading variants have significant prevalence. Once the new variant attains dominance, entropy starts to decrease. The width of the peak in entropy is determined by the speed of this process. For instance, in the subfigure of Supplementary Figure 3. depicting the trends for the United Kingdom, the appearance of either the Alpha or the Delta variants resulted in a much wider peak than that of the emergence of Omicron, in line with a less rapid spread. The collection dates of identified co-infection samples align well with peaks in entropy for both the United States and the United Kingdom.

Other measures of genetic diversity (either the cumulative number of unique lineages up until a time point or the number of concurrently present lineages) seemed to be only moderately affected by the sudden appearance and subsequent swiping of a new variant.

Minor comments:

- Figure 1a: “blue rectangle” – lack legend, not obvious

- Figure 1a: horizontal line at 72, not good visualization, may consider using scaled and piecewise axis.

We thank the Reviewer for pointing out the need for improvement in Figure 1a, the panel have now been slightly modified to make the interpretation more straightforward.

Reviewer #3 (Remarks to the Author):

Pipek et. al Systematic detection of co-infection and intra-host recombination in more than 2 million global SARS-CoV-2 samples

The authors utilize more than 2 million raw SARS-CoV-2 sequencing data files to investigate co-infection and recombination events. Mounting evidence suggests that salutation evolution is at least partly driven by intra-host recombination events. Therefore, detecting and quantifying the extend of intrahost evolution is key to understanding the drivers of rapid SARS-CoV-2 evolution which would likely impact current mitigation strategies.

The authors systematically examine an impressive number of genomes, using raw sequencing data submitted by laboratories throughout the world. The manuscript is clear and well written with significant details of the methodology used. The authors clearly outline the limitations of the analysis conducted.

Major Comments

The authors have conducted an impressive analysis of the countries providing SARS-CoV-2 data that enabled the analysis presented. However, there is no presentation of the breakdown of sequencing technologies, or viral enrichment methodologies used. Could the authors outline at a minimum the distribution of sequencing technologies used to generate the dataset.

We thank the Reviewer for highlighting the lack of insights on this very important technical aspect. Data regarding the sequencing technologies used for analysing co-infection samples is available in Additional Datafile 4. at the repository github.com/csabaiBio/SARSCoV2-coinf. However, due to the immense relevance of the issue, we included the distribution of sequencing platforms and instruments used for the analysis of the samples of various cohorts (all analysed good-quality samples, co-infection samples and samples selected for read-level analysis) as Supplementary Figure 6.

Interestingly, the ratio of all (good-quality) samples sequenced on an Oxford Nanopore instrument (14%) far exceeded the ratio of co-infection samples analysed by long-read sequencing (2%). To further investigate this phenomenon, we calculated the weekly ratio of Nanopore samples in the whole dataset and specifically in co-infection samples for each country and each week whenever at least one co-infection sample was detected. The results of this analysis are presented in Revision Figure 4.

Even though the correlation between the two quantities is undeniable, the actual ratio of co-infection samples sequenced on a Nanopore instrument is frequently lower than the expected value. We hypothesize that this effect might be due to inherent differences in the variant calling pipeline of short vs. long-read sequencing data. Given that Nanopore technologies are more likely to introduce errors during base calling than traditional short-read methods, downstream bioinformatical pipelines treat variations more strictly and are more likely to categorize them as unreliable. For the detection of co-infection samples, a substantial ratio of defining mutations must be present, thus samples analysed with long-read technologies and their subsequent computational processing workflows are not as prone to be identified.

Revision Figure 4. Relationship between the ratio of all good-quality samples sequenced on an Oxford Nanopore instrument and the same of co-infection samples. Each dot represents a single country and a single week. Weeks during which no co-infection samples were detected in the given country are not shown.

Regrettably, metadata describing viral enrichment methodologies is scarce at best and rarely consistent. In an attempt to provide some additional information regarding the protocols used during laboratory procedures, we queried the SRA database⁹ with the Entrez Direct tool¹⁰ for fields "LibraryStrategy", "LibrarySelection" and "LibrarySource" of co-infection samples. To ascertain primer type, we collected all text fields describing experimental details and filtered for ones matching the pattern "primer". We then further scrutinized the resulting strings by searching for various regular expressions that could indicate the utilized primer types and versions in different naming conventions. Whenever this method resulted in no matches, metadata fields containing the word "primer" were additionally inspected manually.

The retrieved results (library strategy, library selection, library source, primer type) have now been added to the already existing table of metadata in Additional Datafile 4.

Further to this point is there any association between detection of Intrahost recombination events within reads and the sequencing technology used? As the authors suggest long read sequencing methodologies may provide an advantage as the longer read lengths are more likely to contain mutations from both parental strains.

We are grateful to the Reviewer for raising this important question. Unfortunately, out of the 118 samples selected for read-level analysis, only two were sequenced on the Oxford Nanopore platform. Our main focus during down-sampling was to maintain the original distribution of variant compositions of co-infection samples in the selected dataset. Additionally, Supplementary Figure 6. demonstrates that the ratio of samples sequenced with long-read technologies was identical for all co-infection samples and for the subset selected for read-level analysis, suggesting that the 118 samples included in downstream investigations serve as a representative sample of the original co-infection cohort.

Nevertheless, as touched upon above, we do agree that a more detailed exploration of how sequencing technology might affect the results would be warranted. Given the extremely low number of available Nanopore samples in the analysed cohort, such nuanced investigations were unavailable to us. For the two indicated samples, no trend for increased detectability of recombination breakpoints was observed, although we refrain from drawing any conclusions based on such limited data. In theory, long-read sequencing could improve the detectability of intra-host recombination in established co-infection samples. However, as discussed above, the inherent features of the Nanopore sequencing workflow makes the identification of such co-infection samples more difficult. Ultimately, while both short and long-read sequencing technologies exhibit unquestionable strengths, they are not without their respective limitations.

More detailed information about the sequencing technology, enrichment methodology, sequencing coverage, sequencing depth, duplication rates and quality metrics would be useful. These quality control parameters are particularly important to validate results from the small number of samples where recombinant reads were detected.

We appreciate the Reviewer's concerns regarding quality metrics and besides the above discussed parameters related to sequencing methodology (library strategy, library selection, library source, primer type), we also included the total base count and the ratio of genomic positions with a sequencing depth of less than 10 in Additional Datafile 4.

During the analysis, we aimed to make sure to only include samples of sufficient quality. To this end, samples were initially filtered by their total base count and the ratio of positions not covered satisfactorily. For co-infection detection, besides extending the list of unique defining mutations to mutually exclusive ones, we used extremely stringent filtering criteria to ensure that selected samples contained at least 80% of the mutually exclusive defining mutations of all of their composing variants. Throughout read-level analysis, reads were further filtered to have a mapping quality and a base quality at the positions of defining mutations of at least 30.

All of these measures were taken to ensure that artifacts are kept at a minimum, although we do agree that both the technical and theoretical limitations of intra-host recombinant detection are immense, and all results should be validated with great care. Thus, we are grateful to the Reviewer for the suggestion to report as many technological details as possible.

Minor Comments

A simple diagram to illustrate the bioinformatic software used by the CoVEO database would be useful to enable the approach to be reproduced.

We are thankful for the Reviewer calling our attention to the opportunity to make our analysis pipeline clearer and more straightforward. Supplementary Figure 7. has now been added to the manuscript which describes the steps of the workflow used during our investigations.

All publicly available data (both raw read sets and subsequent analysis results) can be assessed through the European COVID-19 Data Portal (www.covid19dataportal.org). The variant calling workflow, developed by the Versatile Emerging infectious disease Observatory (VEO, www.veo-europe.eu) consortium ensures through a unified pipeline that sequencing results are comparable across different laboratory protocols and various uploaders. The CoVEO database is a locally stored PostgreSQL database which organizes the analysis results in an easily searchable format. During our investigations, we used python scripts to query the database and visualize the results. The notebooks containing the utilized codes, along with figures and descriptions are available at the github repository [csabaiBio/SARSCoV2-coinf/pipelines](https://github.com/csabaiBio/SARSCoV2-coinf/pipelines).

References

1. Greaney, A. J. *et al.* Comprehensive mapping of mutations in the SARS-CoV-2 receptor-binding domain that affect recognition by polyclonal human plasma antibodies. *Cell Host Microbe* **29**, 463-476.e6 (2021).
2. Greaney, A. J., Starr, T. N. & Bloom, J. D. An antibody-escape estimator for mutations to the SARS-CoV-2 receptor-binding domain. *Virus Evol* **8**, (2022).
3. Starr, T. N. *et al.* Prospective mapping of viral mutations that escape antibodies used to treat COVID-19. *Science (1979)* **371**, 850–854 (2021).
4. Greaney, A. J. *et al.* Complete Mapping of Mutations to the SARS-CoV-2 Spike Receptor-Binding Domain that Escape Antibody Recognition. *Cell Host Microbe* **29**, 44-57.e9 (2021).
5. Li, Q. *et al.* The Impact of Mutations in SARS-CoV-2 Spike on Viral Infectivity and Antigenicity. *Cell* **182**, 1284-1294.e9 (2020).
6. Carabelli, A. M. *et al.* SARS-CoV-2 variant biology: immune escape, transmission and fitness. *Nat Rev Microbiol* (2023) doi:10.1038/s41579-022-00841-7.
7. Bloom, J. D. & Neher, R. A. Fitness effects of mutations to SARS-CoV-2 proteins. *bioRxiv* (2023) doi:10.1101/2023.01.30.526314.
8. Clark, A. *et al.* Global, regional, and national estimates of the population at increased risk of severe COVID-19 due to underlying health conditions in 2020: a modelling study. *Lancet Glob Health* **8**, e1003–e1017 (2020).
9. Sequence Read Archive. <https://www.ncbi.nlm.nih.gov/sra>.
10. Kans, J. Entrez Direct: E-utilities on the Unix Command Line. in *Entrez Programming Utilities Help* (2010).

REVIEWERS' COMMENTS

Reviewer #1 (Remarks to the Author):

The authors responded all my concerns and addressed accordingly.

Reviewer #2 (Remarks to the Author):

The authors made good efforts to address my comments. The revised manuscript is okay with me.

ps. taking log on both axis of Supp Fig 4 can help with the R-sq calculation.

I have no additional comments.

Reviewer #3 (Remarks to the Author):

The authors have sufficiently addressed all my previous comments and suggestions.